# Study on Corrosion Resistance of Stainless-Steel Welded Joints with SnSb8Cu4 and SnZn9

**DOI:** 10.3390/ma16113908

**Published:** 2023-05-23

**Authors:** Jintao Wang, Shengxi Wang, Bo Wang, Xiaohui Han, Yong Liu, Jiehe Ye, Zhan Cheng

**Affiliations:** 1China Academy of Machinery Ningbo Academy of Intelligent Machine Tool Co., Ltd., Ningbo 315000, China; 18342299941@163.com (J.W.); brazecheng@163.com (Z.C.); 2CRRC Qingdao Sifang Locomotive and Rolling Stock Co., Ltd., Qingdao 266000, China; wangshengxi001@163.com (S.W.); 13793237339@139.com (X.H.); 10500032383@crrcgc.cc (Y.L.); yejiehe@163.com (J.Y.)

**Keywords:** stainless-steel car body, soldering, corrosion resistance, microstructure

## Abstract

The use of soldering based on metallurgical bonding, as opposed to conventional rubber sealing, is capable of achieving the firm sealing of stainless-steel subway car bodies, though the corrosion resistance of such joints has rarely been investigated. In this study, two typical solders were selected and applied to the soldering of stainless steel, and their properties were investigated. As indicated by the experimental results, the two types of solder exhibited favorable wetting and spreading properties on stainless-steel plates, and successfully achieved sealing connections between the stainless-steel sheets. In comparison with the Sn-Zn9 solder, the Sn-Sb8-Cu4 solder exhibited lower solidus–liquidus, such that it can be more suitably applied to low-temperature sealing brazing. The sealing strength of the two solders reached over 35 MPa, notably higher than that of the sealant currently used (the sealing strength is lower than 10 MPa). In comparison with the Sn-Sb8-Cu4 solder, the Sn-Zn9 solder exhibited a higher corrosion tendency and a higher degree of corrosion during the corrosion process.

## 1. Introduction

In the field of stainless-steel rail transit [1], connection parts with thicknesses of 0.8 mm, or even 0.6 mm, may need to be sealed. While these parts have low working temperatures and low strength requirements, they are also required to have a beautiful appearance, excellent corrosion resistance, and excellent sealing performance [2]. Sealant has been commonly used [3], though conventional gluing technology has some problems (e.g., poor weather resistance and easy aging) [4].

Zhou Zhang and his team [5] investigated laser–Mig hybrid welding. However, due to the excessive energy of fusion welding, it cannot fully meet the requirements of stainless-steel plate welding for thicknesses over 1 mm, and the welding process requires extremely accurate assembly, which is difficult to achieve in engineering. Moreover, the welding stability is poor, as is the corrosion resistance after welding, increasing the likelihood of undercutting and welding penetration problems [6]. Guoliang Qin and his colleagues [7] explored topics relating to laser brazing–fusion welding. As indicated by their results, the heat input significantly affects the width of the brazing seam, and the laser power required is nearly 1 kW, which is not appropriate for the sealing of thin stainless-steel plates.

Zhang Yan and his team [8] have investigated the brazing of stainless steel, with the precious metal Ag serving as the brazing filler metal, and a high brazing temperature. In addition, extensive research has been conducted on the brazing of stainless steel [9], with an interest in reducing the energy input required. Nevertheless, even when cold metal transition welding [10] is used, several problems remain (e.g., excessive welding heat input and large deformation of the car body after welding) [11], and it is easy to cause sensitization of stainless steel [12,13]. When Cu-based solder is employed for arc brazing, the weld is prone to cracking, air leakage, and other problems caused by the diffusion and migration of copper [14], though the heat input is reduced to a certain extent, and both the wettability and spreadability on stainless-steel plates are high [15].

Brazing is defined as follows: the solder, which has a melting point below that of the base metal, is heated to the melting temperature, and the gap of the base metal is filled with liquid solder to achieve the function of connection and sealing [16]. Soldering here refers to soldering temperatures lower than 450 °C, which especially applies to welding between stainless-steel sheets. Among the commonly used stainless-steel solders, tin-based solder has served as an ideal material for sealing stainless-steel car bodies, due to its low brazing temperature and high metallurgical compatibility [17].

Sn-Pb, one of the most used and recommended solders for brazing stainless steel, has the advantages of a low melting point, a simple manufacturing process, excellent wettability, etc. In the Chicago subway, Sn-Pb-based solders have been used for brazing and sealing, however, with the introduction of the Waste Electrical and Electronic Equipment (WEEE) and The Restriction of the Use of Certain Hazardous Substances in Electrical and Electronic Equipment (RoHS) directives [18], lead, i.e., a material harmful to the human body, has been progressively withdrawn from use. Research into, and development of lead-free tin-based solders has become a focus for scholars in the field of stainless-steel soldering [19]. Zhou Xusheng and his team [20] studied stainless-steel soldering and suggested that the Sn-3.5Ag-0.7Cu alloy exhibits excellent mechanical properties and weldability, good wetting and spreading performance with stainless steel, high strength, and beautiful solder joints; however, the precious metal Ag greatly increases the price of this solder, hindering its application. Numerous scholars began to study Sn-Sb [21] solder and Sn-Cu [22] solder, though the wettability of these solders on stainless-steel plates is poor. On the other hand, the corrosion resistance of solders has not been investigated by these scholars.

Sn-Zn9, a commonly used solder, exhibits the advantages of low eutectic temperature and low manufacturing cost [23], and has been extensively employed in Cu substrate welding [24]. Existing research [25] has suggested that its corrosion resistance in Cu matrixes is poor, though there are few related studies in the field of stainless steel.

At present, Sn-Sb-Cu alloys on the market have excellent antifriction, wear resistance, wetting, and spreading performance [26], have been widely used as coating and bearing bush materials for hydraulic support oil cylinders [27], and their cost is relatively low. Our team has innovatively used an Sn-Sb-Cu alloy as a solder in the field of stainless-steel soldering. After welding, it has excellent forming properties and metallurgical bonding, and its strength can reach above 40 MPa [28]. Therefore, in this study, two types of solder were selected to braze and seal stainless-steel sheets, and the corrosion resistance of the two types of solder after welding was studied in detail.

## 2. Materials and Methods

### 2.1. Sample Preparation

The parent material used is 304 steel plate, and its composition is shown in Table 1. Sn-Sb8-Cu4 and Sn-Zn9 are the solders selected for the test, and their compositions are shown in Table 2 and Table 3, respectively.

The 304 stainless-steel sheets selected for the experiments were 400 mm × 50 mm × 1 mm and 400 mm × 50 mm × 2 mm, respectively. The two plates were soldered by overlapping. Before the experiment, the surface of each workpiece was cleaned with a steel brush, and then acetone, alcohol, and deionized water were used to ultrasonically clean the surfaces to remove oil stains and impurities. The solders selected in the experiment were Sn-Zn9 and Sn-Sb8-Cu4, in the form of wire, 1.6 mm in diameter. The brazing equipment was a continuous fiber laser, model CAMHW1000 (Purchased by Fujian Minan Machinery Manufacturing Co., Ltd., Nan’an, China). The output power of the laser was 0.2 kW, the defocus reached 300 mm, the laser beam inclination was 60°, the spot diameter was 8 mm, and the welding rate reached 2 mm/s. An overlapping form was employed for the test plate, with a lap width of 5 mm and an assembly clearance of 0.2 mm. Furthermore, the filiform solder was preset at the weld site.

### 2.2. Testing and Characterization

#### 2.2.1. Melting Temperature

In the experiment of this study, the solidus and liquidus temperatures of samples were examined through Differential Scanning Calorimetry. Prior to the experiment, the crucible was burned twice, with the purpose of the first burn being to clean the crucible and remove impurities in the crucible. The purpose of the second was to obtain the differential thermal curve of the crucible. In the experiment, 5–8 mg solder alloy samples were weighed each time, and the test was performed under the protection of nitrogen, with the temperature increasing at a rate of 10 °C/min, and the test temperature ranging from ambient temperature to 400 °C.

#### 2.2.2. Shear Performance

To obtain the shear strength of the brazed joint for each type of solder, the sample size was 20 mm × 50 mm by wire cutting, the shear force of the joint was examined using an MTS EXECUTE E44 (Purchased by Jiangsu Zhengrui Taibang Electronic Technology Co., Ltd., Yangzhou, China) electronic universal testing machine, and the tensile rate reached 1 mm/min. Three samples of each solder were tested, and the results were averaged for the final result.

#### 2.2.3. Microstructure Analysis and Corrosion Morphology Observation

Morphology was observed under an optical microscope (OM), a scanning electron microscope (SEM), and an X-ray energy dispersive spectrometer (EDS), and the composition of the brazed joint was analyzed. The etching solution was a 4% nitric acid alcohol solution. After the sample was corroded, it was cleaned with alcohol and then dried. Subsequently, the macroscopic corrosion morphology was observed under a stereomicroscope, the microscopic corrosion morphology was identified through SEM, and the composition of the corrosion area was analyzed.

#### 2.2.4. Analysis of Immersion Test

Each prepared sample was weighed and marked as m_1_. Next, it was soaked in a 3.5% NaCl solution. After soaking for 72 h, 168 h, 240 h, 360 h, and 480 h, the sample was taken out, cleaned with deionized water, blow-dried, and marked as m_2_. Subsequently, the surface corrosion products were removed by ultrasonic cleaning and polishing, dried and weighed, and marked as m_3_. The weight loss of the sample and the weight of the corrosion products were recorded as m_1_-m_3_ and m_2_-m_3_, respectively. Weight loss rate = (m_1_ − m_3_)/(St), where “S” denotes the surface area (1 cm^2^) of the exposed metal of the sample in contact with the corrosive medium, and “t” is the corrosion time. Two repeated tests were performed at each time point to ensure the accuracy of the tests. The final test results were taken as their average.

#### 2.2.5. Analysis of Salt Spray Test

The brazed samples were assigned to two groups and cut into pieces of 20 mm × 50 mm. After marking, the samples were placed into the salt spray test box. The salt spray test conformed to the national standard GB/T 1771-2007 [29]. The temperature in the salt spray test box was 35 °C, and the continuous spraying method was adopted. The corrosive medium was 5% NaCl aqueous solution.

#### 2.2.6. Electrochemical Analysis

In accordance with the national standard GB/T24196-2009 [30], the electrochemical corrosion testing of the solder alloys was carried out with the RST5200F electrochemical workstation. In the test, a three-electrode system was used to measure the corrosion polarization curves of different solders in the 3.5% NaCl aqueous solution by the potentiodynamic scanning method. The reference electrode was a saturated calomel electrode, the auxiliary electrode was a Pt electrode, the potential scanning range spanned −200 mV–200 mV, the scanning rate was 0.5 mV/s, and the test temperature reached 25 °C.

Electrochemical impedance spectroscopy (EIS) employed the same instrument as above, with a working frequency range of 100 kHz to 10 mHz, and the amplitude of the perturbation signal was 10 mV. Before the EIS test, the open-circuit potential should be measured before the system is stable.

## 3. Results and Discussion

### 3.1. Brazing Performance of Solders

Since most solders are multicomponent alloys, the temperature difference of the solid-liquid line in their curves affects brazing quality. With a large temperature difference between the solidus and liquidus of the solder alloy, the low melting point portion of the solder alloy will melt first in the brazing process, and the unmelted high melting point phase of the alloy will hinder the liquid phase of the low melting point phase from spreading on the surface of the base metal, reducing the wettability of the solder. The smaller the temperature difference between the solidus and liquidus of the solder, the faster the solder alloy can spread on the surface of the base metal, such that the performance of brazed joints is enhanced.

Figure 1 and Figure 2 present the solid-liquid line analysis of the Sn-Zn9 solder and the Sn-Sb8-Cu4 solder. An endothermic peak was observed in the DSC melting curve of the Sn-Zn9 solder, with a solidus temperature of 208.18 °C, a liquidus temperature of 221.94 °C, and a solid-liquid interval of 13.76 °C. The melting temperature curve of the Sn-Sb8-Cu4 solder displayed only one endothermic peak, in which the solidus temperature reached 240.06 °C, the liquidus temperature was 248.52 °C and the solid-liquid interval was 8.46 °C. Accordingly, the solid-liquid temperature range of the Sn-Sb8-Cu4 solder was lower, which can be more suitable for low-temperature brazing.

The two solders were heated and melted under the same conditions; the wetting and spreading data are listed in Table 4 and Table 5. The respective solders were sampled at three different positions (marked as 1, 2, and 3). After the solders melted, the melted areas of the solders were connected to calculate the spreading area. As revealed by the experimental results, the wetting and spreading properties of the two solders were close to those of Sn-Pb20 solders (Table 6), and they may both potentially serve as stainless-steel sealing materials in place of Sn-Pb20.

### 3.2. Microstructural Studies of Compounds

Figure 3 presents the macroscopic morphology of 304/Sn-Sb8-Cu4/304 and 304/Sn-Zn9/304 solder joints. The solder fills the whole gap, and the solder joints were well formed, without obvious defects such as inclusions, pores, or cracks.

Figure 4 illustrates the microstructure of the Sn-Sb8-Cu4 solder joint, and Figure 5 shows the microstructure of the Sn-Zn9 solder joint. As depicted in Figure 4, large numbers of small, diamond-shaped, blocky precipitates were uniformly distributed in the Sn-Sb8-Cu4 alloy matrix. Part of the gray matrix phase was the solid solution phase of tin, and in the matrix phase, there were fine, white, granular precipitates, as well as the dispersed blocky precipitates. As depicted in Figure 5, there were obvious rod-shaped aggregation pits in the Sn-Zn9 solder, except for in the gray Sn base.

In order to explore the microstructural distribution and composition of the two solders after brazing in more depth, in this experiment, the morphologies and compositions of the brazed joints were observed by SEM and EDS. As depicted in Figure 6 and Figure 7, the diamond-shaped square components in the Sn-Sb8-Cu4 alloy solder were mainly composed of Cu and Sn, and the needle-shaped components were mainly composed of Sn and Sb. The aggregated part of the Sn-Zn9 alloy solder was mainly composed of Zn.

The standard for quantitative analysis by EDS is GB/T17359-2012 [31], which recommends an accelerating voltage of 10 kV–25 kV. However, the K-line of the Cr and Cu contained in the solder cannot be fully excited at 10 KV, meaning that quantitative analysis cannot be carried out, therefore the accelerating voltage used in this experiment was 20 KV.

As depicted in Figure 8, the EDS analyses of the small diamond-shaped blocks and small white needles in the Sn-Sb8-Cu4 solder indicated that the element compositions at points A and B were the same, and the molar ratios of Sn, Cu, and O were x (Sn)/x (Cu)/x (O) = 46.80/54.20/21.22 and x (Sn)/x (Cu)/x (O) = 49.92/55.70/19.88, respectively. The molar fraction ratios of points A and B were similar, such that the needle precipitates in the two places were in the same precipitation phase. Given the effect of the tin-based solid solution on the content of Sn, calculations suggest that the diamond-shaped precipitates were in the Cu6Sn5 phase. The diamond-shaped precipitates at points C and D were mainly Sn and Sb, and the molar ratios of Sn, Sb, O were x (Sn)/x (Sb)/x (O) = 47.87/30.91/6.16 and x (Sn)/x (Sb)/x (O) = 48.63/34.30/5.89, respectively. In addition, their oxygen contents were relatively low.

As revealed by the analyses of the pits in the Sn-Zn9 solder, the element composition was almost entirely Sn and O at point C, and the compositions at points A and B were similar to each other, with molar ratios of x (Zn)/x (O) = 5.02/54.55 and x (Zn)/x (O) = 3.35/37.38, respectively.

The above results suggested that during the solidification process, the Zn element condensed to form a coarse, zinc-rich phase, which can be easily corroded because of its high potential. Moreover, the potential difference between the Zn element and the Sn element is large, and the mixed distribution of these two elements in the solder means that a primary battery reaction may occur easily. Thus, the alloy surface was in an activated state, and the non-metal surface was susceptible to corrosion. Due to the low Zn content, the anode area was significantly smaller than the cathode area, and the anode area was therefore seriously corroded, resulting in a narrow and deep corrosion pit.

However, in the Sn-Sb8-Cu4 solder, the potentials of the elements were similar, and the β-SbSn and Cu6Sn5 phases were hard and stable, such that it was difficult for the corrosive medium to enter and corrode them. In addition, as the electrode potentials of the respective phases were close, it was difficult for corrosion to occur.

### 3.3. Corrosion Weight Loss Analysis

Figure 9 and Figure 10 present the weight loss test results of the Sn-Sb8-Cu4 solder and the Sn-Zn9 solder in the 3.5% NaCl solution. After being corroded in the 3.5% NaCl solution for 480 h, the weight loss and weight loss rate of the Sn-Zn9 solder reached 57.2 mg and 11.91 μg/cm^2^·h, respectively. However, after 480 h of corrosion in the 3.5% NaCl solution, the weight loss and weight loss rate of the Sn-Sb8-Cu4 solder only reached 14.7 mg and 3.06 μg/cm^2^·h, respectively. The corrosion rate of the Sn-Zn9 solder in the 3.5% NaCl solution was four times that of the Sn-Sb8-Cu4 solder.

### 3.4. Analysis of Salt Spray Tests

The brazed joints of stainless steel may be exposed to all types of environments, and may come in direct contact with all types of media during their use. Notably, when the joints are employed in high-speed rail trains, the brazed joints will be exposed to the atmospheric environment, and the water vapor, salt ions, and impurities in the environment will be adsorbed onto the surfaces of the joints, such that corrosion of the joints may occur. The corrosion resistance of brazed joints of Sn-Sb8-Cu4 and Sn-Zn9 solders was investigated using the neutral salt spray test. In the test, the corrosion medium was 5% NaCl salt spray, and the corrosion time reached 672 h. The changes in shear strength of the brazed joints during the test time is presented in Figure 11. As depicted in Figure 11, with the salt spray test, the shear strength of brazed joints tended to be decreased. However, during the process of corrosion, the shear strength of the Sn-Sb8-Cu4 solder was decreased to a lesser extent than that of the Sn-Zn9 solder.

### 3.5. Electrochemical Analysis

Figure 12 illustrates the polarization curves of the Sn-Zn9 and Sn-Sb8-Cu4 solders in a corrosive solution. As depicted in the figure, the self-corrosion potential, Ecorr, the self-corrosion current density, icorr, and the linear polarization resistance of the Sn-Sb8-Cu4 solder were −495.5 mV, 5.4 nA·cm^−2^, and 214.7 kΩ·cm^2^, respectively, while the Sn-Zn9 solder values were −1107.6 mV, 603 nA·cm^−2^, and 25.3 kΩ·cm^2^, respectively. These results suggested that the self-corrosion potential of the Sn-Sb8-Cu4 solder was higher than that of the Sn-Zn9 solder, and the corrosion tendency was lower. The self-corrosion current density of the Sn-Sb8-Cu4 solder was lower, suggesting that its corrosion rate was reduced. The linear polarization resistance of the Sn-Sb8-Cu4 solder was greater than that of the Sn-Zn9 solder, and the corrosion resistance was improved.

When the Sn-Zn9 alloy was placed in a NaCl solution, it showed a higher tendency towards galvanic corrosion than the Sn-Sb8-Cu4 solder due to the accumulation of Zn, such that its corrosion driving force was enhanced. In comparison to the Sn phase, the Zn-rich phase exhibited a higher reversible potential. Under the effects of a corrosive medium, the Zn-rich phase was preferentially corroded as the anode of the corrosion cell. In contrast, the hard point phases (the β-SbSn and Cu6Sn5 phases) in the solder in the Sn-Sb8-Cu4 alloy were uniformly distributed and were all stable phases. Therefore, the corrosive medium could not easily enter and corrode these phases, making the Sn-Sb8-Cu4 alloy difficult to corrode.

Figure 13 and Figure 14 present the electrochemical impedance spectroscopy of the brazed samples of the two solders. As indicated by the Bode plot, the impedance modulus of the Sn-Zn9 solder was lower than that of the Sn-Sb8-Cu4 solder at the low-frequency band of 0.01 Hz, suggesting that its corrosion resistance was poor, its barrier effect to the medium was poor, and its corrosion tendency was higher than that of the Sn-Sb8-Cu4 solder.

## 4. Conclusions

In this study, the performance of solders employed for the purposes of sealing and brazing stainless steel was examined, and the corrosion resistance was systematically analyzed. Thus, this study can provide a useful reference for future researchers interested in identifying stainless-steel solders with better performance than the Sn-Zn9 solder. The conclusions of this study are as follows:The Sn-Zn9 and Sn-Sb8-Cu4 solders are capable of replacing the sealants used to seal the stainless-steel car body. After sealing, the surface of the brazing seam may be well formed, smooth, and continuous, with almost no deformation after welding; the surface does not change color.The Sn-Zn9 and Sn-Sb8-Cu4 solders have exhibited excellent wetting properties, spreading properties, and sealing strength in this experiment on stainless-steel sealing soldering.In comparison with the Sn-Zn9 solder, the Sn-Sb8-Cu4 solder exhibits better corrosion resistance in immersion, salt spray, and electrochemical experiments.

## Figures and Tables

**Figure 1 materials-16-03908-f001:**
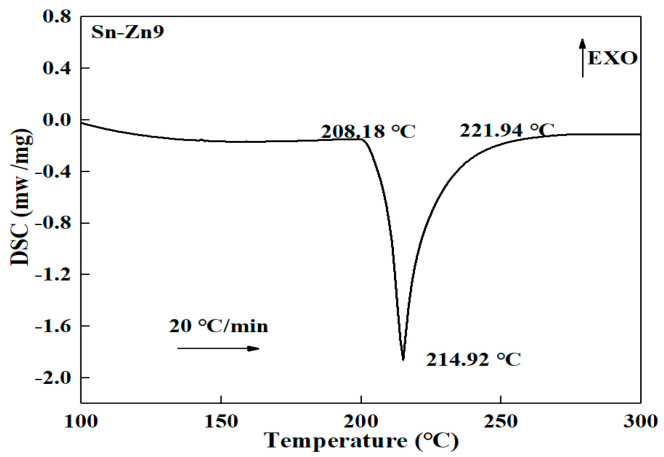
DSC curve of the Sn-Zn9 solder.

**Figure 2 materials-16-03908-f002:**
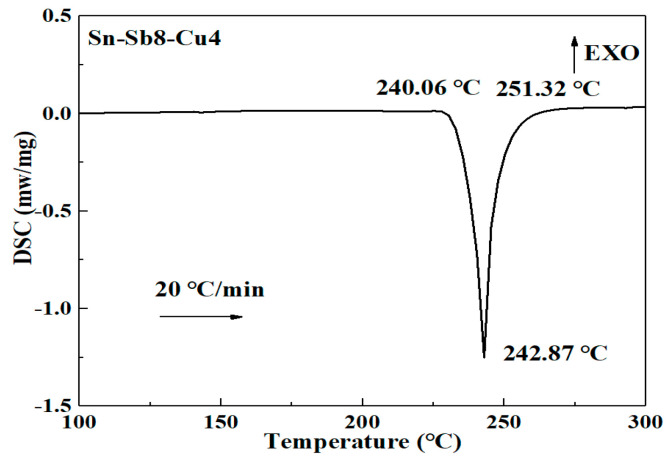
DSC curve of the Sn-Sb8-Cu4 solder.

**Figure 3 materials-16-03908-f003:**
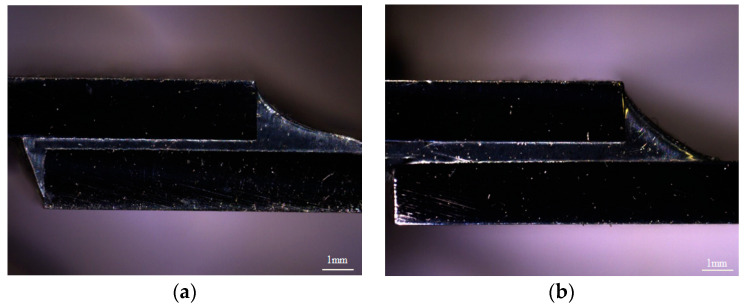
Morphology of the two solders after brazing: (**a**) Sn-Zn9 and (**b**) Sn-Sb8-Cu4.

**Figure 4 materials-16-03908-f004:**
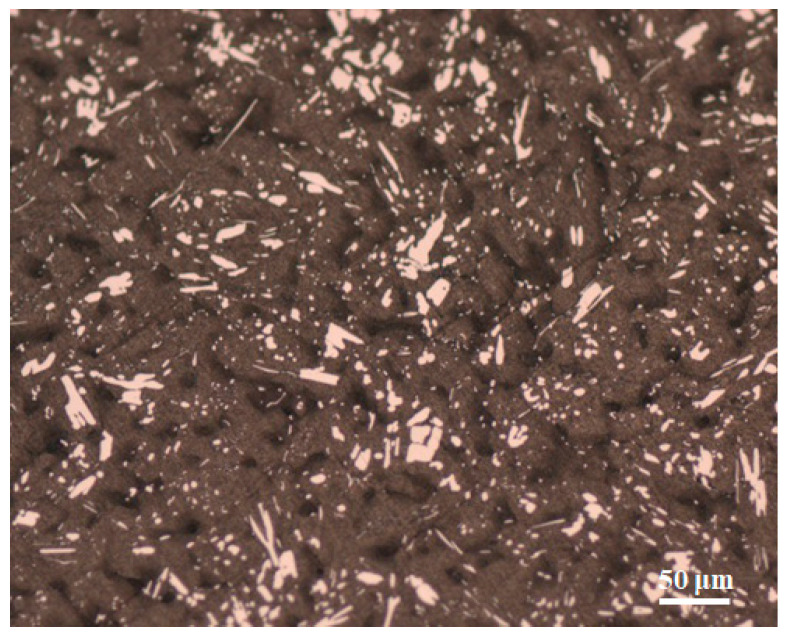
Microstructure of the Sn-Sb8-Cu4 solder.

**Figure 5 materials-16-03908-f005:**
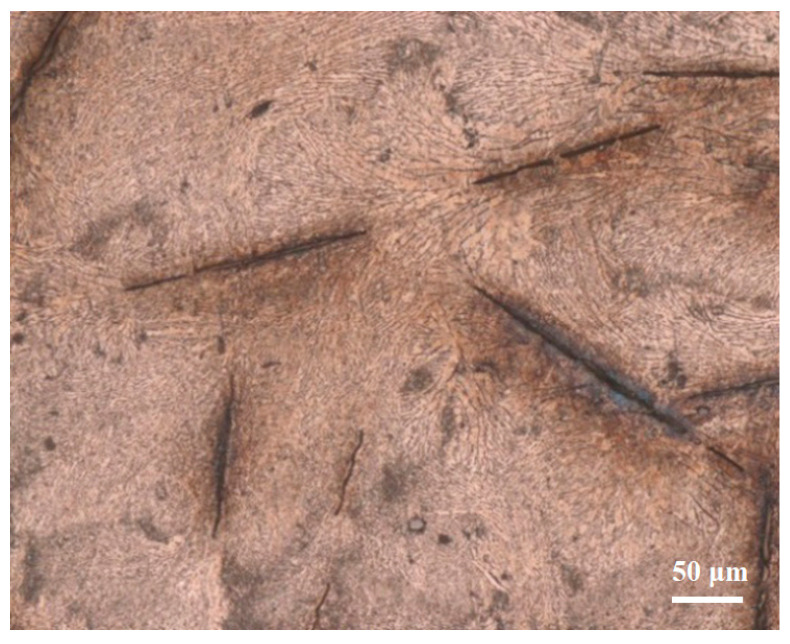
Microstructure of the Sn-Zn9 solder.

**Figure 6 materials-16-03908-f006:**
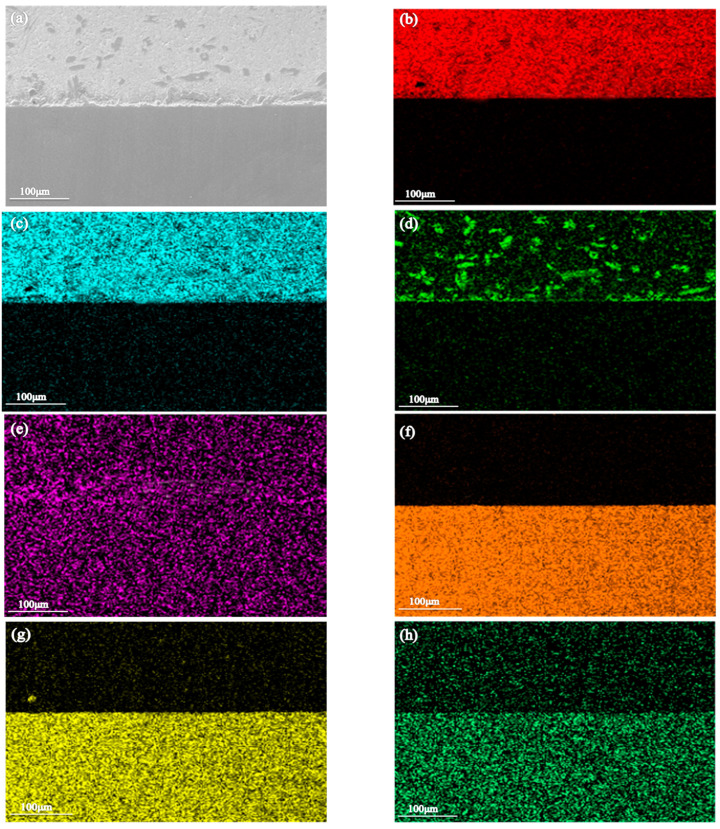
SEM image and EDS element distributions of the Sn-Sb8-Cu4 alloy brazing seam area. (**a**) Microstructure of brazing seam and distributions of: (**b**) Sn, (**c**) Sb, (**d**) Cu, (**e**) O, (**f**) Fe, (**g**) Cr, and (**h**) Ni.

**Figure 7 materials-16-03908-f007:**
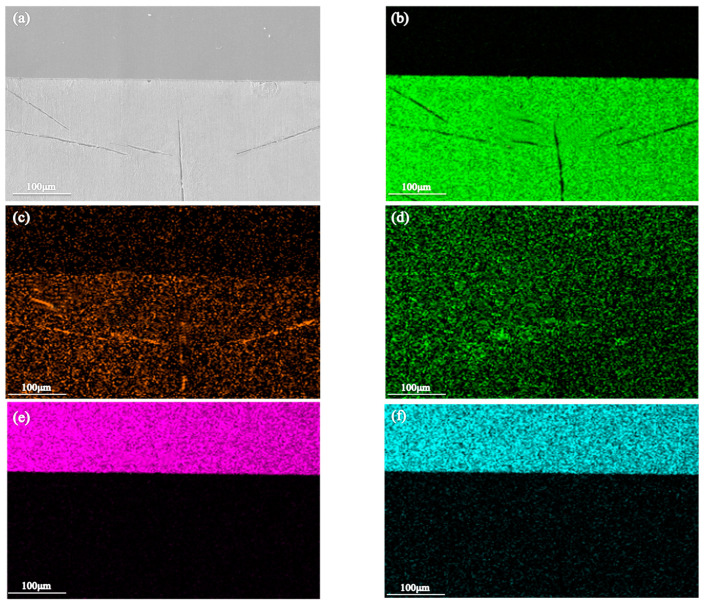
SEM image and EDS element distributions of the brazing seam area of the Sn-Zn9 alloy. (**a**) Microstructure of brazing seam and distributions of: (**b**) Sn, (**c**) Zn, (**d**) Fe, (**e**) Cr, and (**f**) Ni.

**Figure 8 materials-16-03908-f008:**
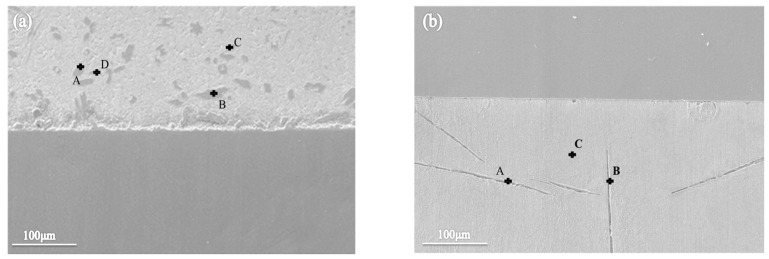
Scanning microstructure diagrams of two types of filler metal: (**a**) Sn-Sb8-Cu4, and (**b**) Sn-Zn9.

**Figure 9 materials-16-03908-f009:**
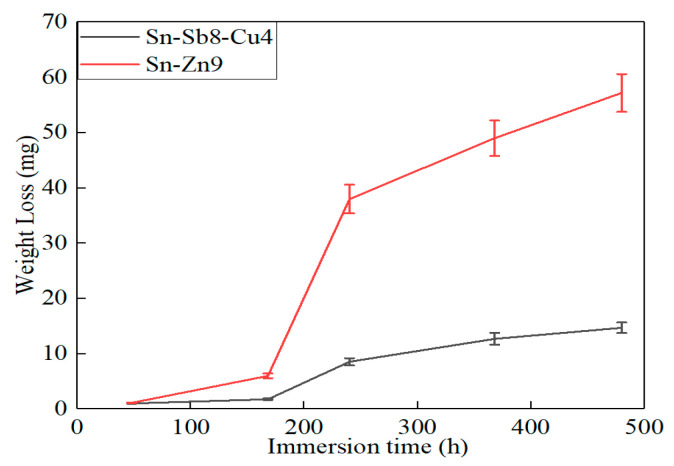
Weight loss curve of the two solders during the immersion test.

**Figure 10 materials-16-03908-f010:**
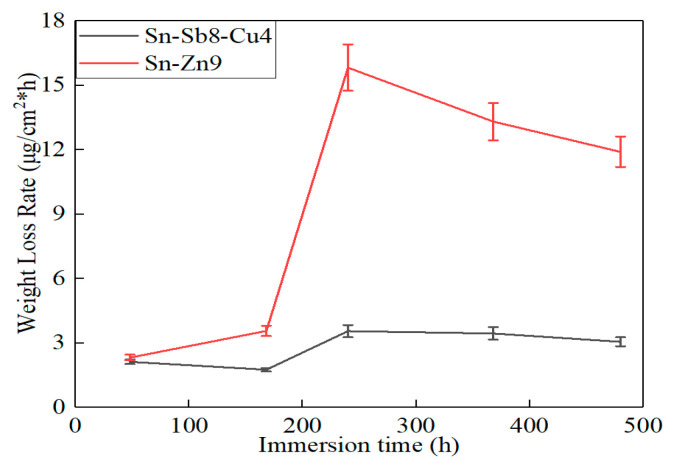
Weight loss rate curve of the two solders during the immersion test.

**Figure 11 materials-16-03908-f011:**
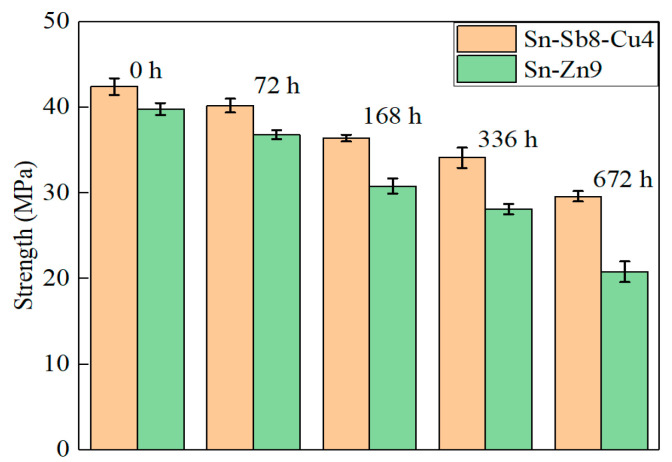
Changes in shear strength of brazed joints after the neutral salt spray test.

**Figure 12 materials-16-03908-f012:**
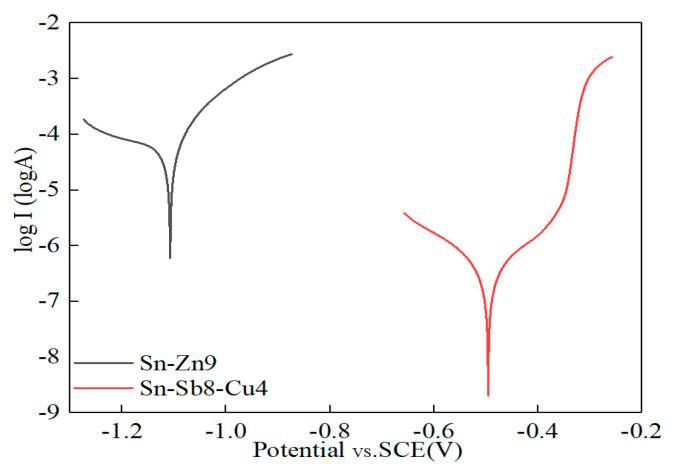
Self-corrosion potential diagram.

**Figure 13 materials-16-03908-f013:**
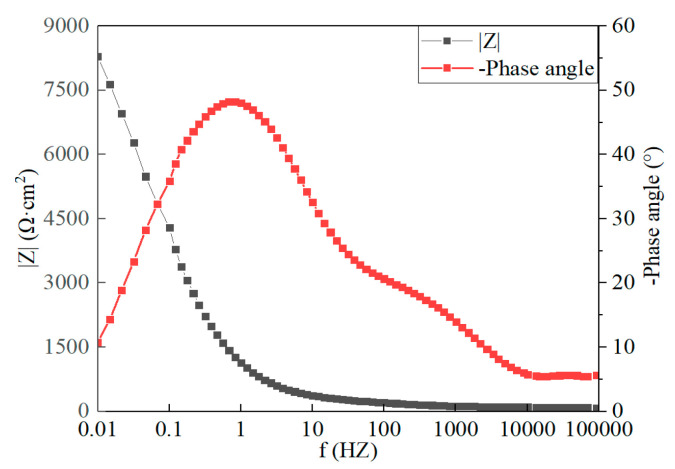
Bode and Nyquist plots of the Sn-Sb8-Cu4 solder.

**Figure 14 materials-16-03908-f014:**
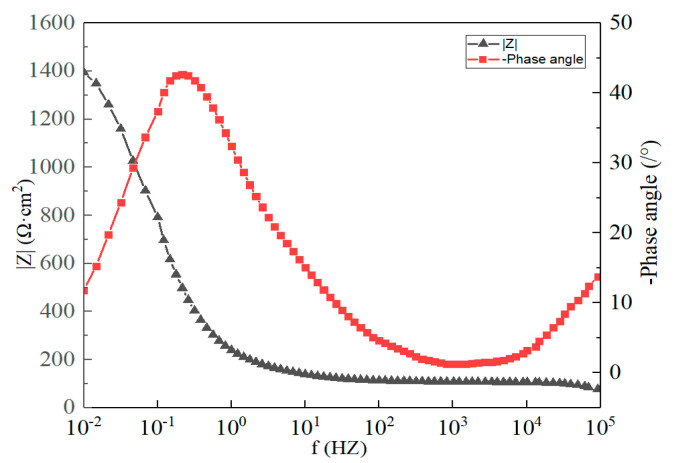
Bode and Nyquist plots of the Sn-Zn9 solder.

**Table 1 materials-16-03908-t001:** Chemical composition of 304 stainless steel (wt.%).

C	P	Si	Mn	Ni	Cr	Fe
0.07	≤0.045	≤0.030	2.00	8–11	18–20	residual amount

**Table 2 materials-16-03908-t002:** Chemical composition of Sn-Sb8-Cu4 stainless steel (wt.%).

Sb	Cu	Fe	As	Al	Sn
7.72	3.81	≤0.005	≤0.004	0.013	residual amount

**Table 3 materials-16-03908-t003:** Chemical composition of Sn-Zn9 stainless steel (wt.%).

Zn	Cu	Fe	As	Al	Sn
8.82	0.01	≤0.005	≤0.004	0.013	residual amount

**Table 4 materials-16-03908-t004:** Wetting and spreading area of the Sn-Zn9 solder.

Number	Temperature/°C	Time/min	Area/mm^2^
1	280	3.5	44.61
2	280	3.5	46.31
3	280	3.5	48.14

**Table 5 materials-16-03908-t005:** Wetting and spreading area of the Sn-Sb8-Cu4 solder.

Number	Temperature/°C	Time/min	Area/mm^2^
1	280	3.5	48.53
2	280	3.5	44.80
3	280	3.5	42.66

**Table 6 materials-16-03908-t006:** Wetting and spreading area of the Sn-Pb20 solder.

Number	Temperature/°C	Time/min	Area/mm^2^
1	280	3.5	51.13
2	280	3.5	49.89
3	280	3.5	51.68

## Data Availability

The data could be obtained from the corresponding author.

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
