# Peer review of "Study on Corrosion Resistance of Stainless-Steel Welded Joints with SnSb8Cu4 and SnZn9"

_materials, 2023, doi:10.3390/ma16113908_

Round 1

Reviewer 1 Report

Below are a few remarks:

Not a single result obtained by the authors is compared to data from the literature!?! Can the shear strength data be compared with those of actual materials used for welding stainless steel rails? The readers would like to know if, effectively, SnSb8Cu4 and SnZn9 are potential candidate materials.

Also the authors cited 16 references but only one that is not from Chinese researchers!... Here is one reference from the journal Materials :

Zaneta Gerhatova, Paulína Babincova, Marian Drienovsky, Matej Pasak, Ivona Cernickova, Libor Duriska, Robert Havlík and Marian Palcut, ‘Microstructure and Corrosion Behavior of Sn–Zn Alloys’, Materials 2022, 15, 7210. https://doi.org/10.3390/ma15207210

Definition of weightlessness : condition experienced while in free-fall, in which the effect of gravity is canceled by the inertial (e.g., centrifugal) force resulting from orbital flight. The term zero gravity is often used to describe such a condition. The authors just did immersion corrosion tests.

Lines 180-193 : There are no indications of the SEM EDS analysis conditions, so we can have doubt on the results of the EDS analyses. The SnZn9 solder should have an eutectic structure. Thus the dark phase should be Zn rich with low solubility of Sn). It is surprising that the authors obtained a composition ‘“A” and “B” are similar, with 192 x(Zn) /x(O) = 5.02/54.55 and x(Zn) /x(O) = 2.35/37.38’. The spatial resolution is dominated by the size of the interaction volume? This interaction volume is determined by the accelerating voltage of the SEM. It is recommended to work with a low accelerating voltage, such as 5 kV.

Line 78 : ‘In the experiment, 5-8g solder alloy samples were ...’ 5-8 grams seem to be a lot! Would not is be 5-8 mg?

English

Citation of references

Line 24 : ‘Wen et al. [4] ...’ the first name of Mr or Ms Wen is not necessary, but it is important to mention that this article was written with colleagues).

Line 24 : ‘Zhao et al. [5] ...’ the same as above!

Line 38 : What are WEEE and RoHS? Please define acronyms !

Line 45 : ‘Chen and Li [13]...’ : when only 2 authors, we should mention both name.

Line 56 : MPa

Lines 96-98: These are instructions! Probably given by your adviser! Please correct!

Lines 106-107 : These are instructions, too! Please correct!

Lines 170-175 : Please use the past tense everywhere in your text.

References:

Sometime only the initials of the first names are given, sometime the full first name is given...

1. Cheng G, Cheng P, Chao MA. Application of Spot Welding in the Manufacture of Stainless Steel Rail Transit Vehicle. Urban Mass Transit, 289 2019, 2, 4-6. Doi: 10.16037/j.1007-869x.2019.02.026 290

2. Liu C, Ji HU. Buckling Analysis of Rail Transit Stainless Steel Carbody Based on American Standards. Urban Mass Transit, 2018. 291 Doi: CNKI:SUN:GDJT.0.2018-02-004 292

3. Long Weimin, Sun Huawei, Qin Jian, et al. Application of brazing technology in high-speed train manufacturing. Electric 293 welding machine, 2018, 48 (03): 25-31. Doi: 10.7512/j.issn.1001-2303.2018.03.04 294

4. Wen Long, Zhang Ruoyan, Shi Yan. Study on Corrosion Resistance of 304 Stainless ...

Why can’t the authors follow the journal recommendations?

Line 301 : 7. Setia P , Anand A , Venkateswaran T , et al. Effect of Heat Treatment on the Microstructure Evolution and Sensitization Be-301 havior of High-Silicon Stainless Steel. J.Mater.Eng.Perform, 2020, 10, 1-11. Doi:10.1007/s11665-020-05060-w

Please cite all the author names! Prince Setia, Ayush Anand, T. Venkateswaran, K. Thomas Tharian, Sudhanshu S. Singh, K. Mondal, and Shashank Shekhar

Reviewer 2 Report

The manuscript is devoted to the corrosion resistance of SnSb8Cu4 and SnZn9 welds of stainless steel. The topic is valuable for industrial practice, and the work may be of interest to scientific community.  Therefore I believe the paper is worth publishing - however manuscript needs major revision.

My specific comments and suggestions, in the order in which they appear in the manuscript:

1. Editing of English language and style required.

2. lines 24-25 
"Wen Long [4] studied YAG laser gas shielded welding, Zhao Ruirong [5] studied high purity argon gas shielded welding." 
The manuscript lacks information about the effect of the research of the cited authors.

3. Information mismatch 
On line 100 it says that the corrosive medium is 5% NaCl, and on line 226 it says 3.5% NaCl.

4. line 120 
"the potential scanning range is -200 mV ~ 200 mV" 
I suppose that the potential range from -200 to 200 mV refers to the value of the corrosion potential (open circuit potential). This should be written in the text of the manuscript.

5. line 123 
"disturbance voltage of 10 mV" 
In the field of electrochemistry, the term "amplitude of perturbation signal" is used.

6. title of Table 1 
delete "SnZn9" before the word "Wetting"

7. title of section 3.2. "Organizational analysis" is inappropriate

8. lines 170-175 
Citation to Figures 6 and 7 should be written in this paragraph.

9. Figures numbering 
Two of the figures on page 8 are numbered as 9 (instead of 9 and 10). In effect, the numbering in the titles of next figures is reduced by one and is inconsistent with the figures descriptions in the text of the manuscript.

10. Figures on page 8 
a) The corrosion rate is proportional to the weight loss, so the shapes of the graphs in both figures should be identical. Why isn't it? 
b) The deviation/error bars should be marked on the graphs.

11. line 246 
"Zn-rich phase will preferentially corrode as the anode of corroded battery" 
The term "corrosion cell" is used, not "corroded battery".

Reviewer 3 Report

The work was done on a topical issue - the formation of permanent connections for railway transport. Ensuring functionality (corrosion resistance and durability) along with bond strength is an important technical challenge.

1. The introduction deals with the problem of forming a non-detachable hermetic connection of a thin-sheet stainless steel product. The problem is formulated and the way to solve it is determined. A brief overview of the field of soldering and welding using different techniques and solder material is given.

The review uses publications mainly published in the last 5 years. At the same time, there are few references - only 16 sources.

2. The second section describes in detail the materials and research methods. The methods chosen are modern and adequately suited to the chosen research topic.

3. The third section describes the results obtained and discusses them.

4. The fourth section contains conclusions based on the new results obtained.

Remarks

1. The text sometimes uses different fonts.

2. item 1.4 on page 8 is a typo.

3. Abstract is worth writing a little more detailed about the work done and the new results achieved.

4. I recommend to slightly expand the review of the literature. 16 citations are not enough for a modern scientific publication.

5. From the content of tables 1 and 2 it is not entirely clear what is the spreading area and what is the wetting area. It is also not clear what is included in the column «Number» - is it the number of the sample?

6. Section 3.2. named incorrectly. It mainly contains data on structural studies. In this regard, it would be more correct to name the section as microstructural studies of compounds.

7. The numbering of the figures is not correct starting from the 9th figure.

8. When discussing the results, the authors do not refer to known literature data and do not conduct a comparative analysis.

Round 2

Reviewer 1 Report

Would it be possible to use the conventional notation of alloys ?

For example : SbSn corresponds to the phase Sb50Sn50 the percentage corresponding to atomic percent. The authors refer to the beta-SbSn phase page 285 as well as to the Cu6Sn5 phase in the solder Sn-Sb8-Cu4 alloy. Indeed in the notation Sn-Sb8-Cu4, the numbers indicated iin the composition correspond to weight percent !

Indeed the autors wrote line 164 : Sn-Zn solder !

The reader assumes that in the title, the proper denomination should be  Sn-Sb8-Cu4 and Sn-Zn9 since the concentration of Zn in the Sn-Zn9 is approximately 9 weight percent.

Materials and methods :

Could the authors give the chemical compositions of the solders ?

English

Line 45 : ‘cracking’ (... more prone to cracking, …)

Line 61 : ‘... has progressively been withdrawn...’ been was missing !

Line 69 : The sentence should stop at :’ ‘...stainless-steel plates is poor.’ And could the authors rewrite the sentence : ‘On the other hand, these scholars' research paid little attention to the corrosion resistance of solders’

Line 226 : ‘analyses’ plural form because several analyses...

Line 230 : ‘...which can easily be corroded because of its...’ instead of ‘... which can be easy to be corroded for its...’

Line 232 : ‘...and it can be easy to form...’ please rewrite in proper English

Line 255 : ‘3.4 Analysis of salt spray tests’ Hopefull the authors performed more than one salt spray test !

Line 304 : not elucidated ! plese find an appropriate verb.

Reviewer 2 Report

In the revised manuscript the authors addressed my comments and recommendations, and answered my questions. I believe the manuscript has been sufficiently improved and it can be accepted in present form.
